# Reduction of Cancer-Induced Thrombocytosis as a Biomarker of Improved Outcomes in Advanced Gastric Cancer

**DOI:** 10.3390/jcm11051213

**Published:** 2022-02-24

**Authors:** Kamil Konopka, Paulina Frączek, Maciej Lubaś, Agnieszka Micek, Łukasz Kwinta, Joanna Streb, Paweł Potocki, Piotr J. Wysocki

**Affiliations:** 1Department of Oncology, Jagiellonian University Medical College, 31-007 Cracow, Poland; 2Department of Medical Oncology, University Hospital in Cracow, 30-688 Cracow, Poland; 3Department of Nursing Management and Epidemiology Nursing, Jagiellonian University Medical College, 31-007 Cracow, Poland

**Keywords:** gastric cancer, platelets, chemotherapy

## Abstract

Background: Interplay between non-specific inflammatory reaction and tumor microenvironment in gastric cancer (GC) can be measured indirectly by assessing fluctuations in concentration of platelets. Cytotoxic chemotherapy affects these morphotic elements directly by inducing myelosuppression. It was hypothesized that chemotherapy not only directly affects malignant cells, but also through immunomodulation related to myelosuppression. Methods: Metastatic GC patients (N: 155) treated with chemotherapy +/− trastuzumab were enrolled in this retrospective study. Platelet pretreatment concentration (PLT-count) and the deepest level of platelet reduction, as well as other inflammatory and general confounders were collected in the first 12 weeks of treatment (PLT-red). Martingale residuals were used to visualize the relationship between PLT-count, PLT-red, and overall survival (OS). Multiple multivariate Cox regression models were built to assess the impact of platelet reduction on OS and progression-free survival (PFS). Results: Reduction of PLT (PLT-red) to 60% of baseline concentration was associated with improved survival rates (HR = 0.60, *p* = 0.026 for OS and HR 0.56, *p* = 0.015 for PFS). Cross-classification into four groups based on PLT-count (high vs low) and PLT-red (high vs low) showed significantly worse survival rates in both high PLT-count (HR = 3.60, *p* = 0.007 for OS and HR = 2.97, *p* = 0.024 for PFS) and low PLT-count (HR = 1.75, *p* = 0.035 for OS and HR = 1.80, *p* = 0.028 for PFS) patients with insufficient platelets reduction. Conclusion: Thrombocytosis reduction represents a novel, clinically important, prognostic factor for OS and PFS in patients with stage IV GC.

## 1. Introduction

The oncology landscape is constantly evolving and there is no doubt that gastric cancer treatment will no longer be based solely on cytotoxic chemotherapy [1,2]. However, until then, optimization of cytotoxic chemotherapy is of the most importance. The long-standing pursuit has been made to find the link between systemic inflammatory reactions and the effectiveness of cancer treatment [3,4].

Several studies have shown that there is a correlation between baseline systemic inflammation measured by neutrophil-to-cell ratio (NLR) and overall survival (OS) in the context of both cytotoxic chemotherapy [5] and immunotherapy [6]. While the role of NLR appears to be solid if not explained, the importance of another morphotic element, platelets, is more ambiguous [7]. Several studies have shown that platelet-based markers, such as the platelet-to-lymphocyte ratio (PLR) [8] or platelet-derived weight (PDW) [9], could provide prognostic information. Even more important than finding accurate prognostic factors is finding good predictive factors to steer the course of treatment that is already on the way. The detection of early signs of the futility of treatment may guide the early modification of systemic therapy. The characteristic and typical adverse event of cytotoxic chemotherapy is myelotoxicity [10], and some studies showed that treatment-induced neutropenia [11] or thrombocytopenia [12] could predict a good response to therapy. Overall, it remains unclear whether such cytopenias might indeed be helpful in tailoring patient-oriented treatment.

We hypothesized that chemotherapy-mediated reduction of initially increased platelet count could be used as an early biomarker that predicts response to systemic treatment. Platelets are better suited for this task because of variability of the neutrophil count resulting from the nature of neutrophil progenitor cells. Furthermore, unlike in the case of low-grade thrombocytopenia, chemotherapy must be delayed in most patients, demonstrating neutropenia (≥G2 CTCAE) due to the risk of subsequent febrile neutropenia [13]. Furthermore, in many patients with neutropenia, G-CSF administration is required, which can also result in clinically significant neutrocytosis [14].

To evaluate the prognostic and predictive role of thrombocytosis and the reduction in treatment-induced PLT count, we analyzed the extent of a decrease in the number of circulating PLT during the first 12 weeks of treatment, which is the usual time point for evaluating the initial response to systemic therapy in clinical practice.

## 2. Materials and Methods

The retrospective analysis included data on 155 patients with advanced gastric cancer (GC) treated systemically in the Department of Clinical Oncology, University Hospital in Cracow between September 2013 and December 2019. The data was censored on 31 December 2020. Patients were eligible for analysis if at least one cycle of palliative intravenous chemotherapy had been administered and full longitudinal records of blood morphology were available. The chemotherapy regimen administered was a standard of care at a given time and was adapted to the patient’s performance status, HER2 expression, and available therapeutic guidelines. The neutrophil to lymphocyte ratio (NLR) was calculated by division of the absolute neutrophil and lymphocyte counts. Platelet reduction (PLT-red) was calculated as a ratio of the lowest PLT count between the 2nd and 12th week divided by the initial platelet concentration (PLT-count). Overall survival (OS) was defined as the time from the initiation of the first palliative chemotherapy to death. Progression-free survival (PFS) was defined as the time from the beginning of the first palliative chemotherapy to progression or death. The intensity of the dose was calculated as the ratio between the number of cycles of chemotherapy administered, divided by the maximum possible in a given time. Chemotherapy regimens were classified into four categories according to the number of drugs administered (single agent regime, double agent regime, triple agent regime, and trastuzumab-based regimen).

### Statistical Analysis

The baseline characteristics of the study sample were presented in two groups of patients who achieved or did not achieve a meaningful level of PLT reduction. Categorical variables with frequencies and percentages were reported and continuous variables were characterized by medians and interquartile ranges due to the high right skewness of all investigated parameters. Nonparametric tests were used to compare medians between two or more independent groups (Wilcoxon rank sum test and ANOVA rank Kruskal-Wallis H test). The functional form of the relationship between continuous variables and the log hazard ratio was tested with martingale residuals [15]. The Kaplan–Meier method was used to estimate survival functions, and a comparison of them between two or more independent groups was performed applying a log-rank test. Multiple Cox regression models were fitted to assess the influence of the two main exposure variables PLT-count and PLT-red, as well as the impact of covariates on time to death or disease progression. NLR at baseline, dose intensity, ECOG, and type of chemotherapy were considered potential confounders based on previous evaluation [16]. The proportional hazard assumption in all multivariable models was tested with visual inspection of Schoenfeld residuals and was formally complemented with omnibus chi-squared goodness of fit tests, relating failure time to covariate values. In case of violation of this assumption, a different baseline hazard function was introduced. Dichotomization of inflammatory markers or their reduction was performed using cut-off points determined by displaying diagnostic graphs of Cox models. To fulfill the assumption of linearity, the logarithmical transformation (with a base 2) was used. The analyses were performed in R software (Development Core Team, Vienna, Austria, version 4.0.4). All tests were two-sided and statistical significance was defined as *p* < 0.05. No correction was applied for multiple statistical tests, due to the exploratory nature of the study.

## 3. Results

The final analysis consisted of 105 patients. Fifty patients were excluded due to the lack of complete longitudinal data.

The maximum level of platelets reduction was usually observed in the 7th week of treatment (Me: 7, IQR: 5–10 weeks).

To visually assess the relationship between OS and platelet-based variables and to check the functional form of exposure, which should be used in analysis, we plotted two charts with martingale residuals computed from the null COX regression and modelled them as a function of either PLT-count or PLT-red (Figure 1).

We detected a linear positive trend in association with PLT-red up to a value of 60% reflecting a worsening survival prognosis, followed by stabilization with achievement of the plateau afterwards. Analogical visual inspection of the relationship with the PLT-count revealed that crossing the 400 × 10³/µL level was associated with a gradual increase in martingale residuals and, consequently, a continuous decrease in the average duration of survival. The analysis in relation to PFS (Appendix A) showed similar patterns of the relationship with roughly the same cut-off points (60% for PLT-red and 400 × 10³ /µL for PLT-count), therefore, these values were used as thresholds for dichotomization. On the other hand, analyzing PLT-red as a continuous variable was admissible after logarithmical transformation with a base of 2.

Kaplan–Meier survival analysis was performed to compare the survival time of patients with high PLT count (*n* = 28) and low PLT count (*n* = 77). The median OS in the low PLT count group was 9.5 months (95%CI: 8.5–14) and was significantly higher than in the high PLT count group 8.62 (95% CI: 6.0–13.5) (Figure 2). A log rank test depicted the significance of differences in the survival distributions as well (χ^2^(1) = 4.8, *p* = 0.028).

The reduced and non-reduced PLT groups differed significantly in terms of PLT-count and PDW, but not in lymphocytes and neutrophil concentration, or NLR ratio. There were no differences in chemotherapy dose intensity between the groups; however, compared to PLT-reduced patients, the PLT-non-reduced subsample more frequently received single agent chemotherapy and simultaneously less frequently two-drug chemotherapy. Single agent chemotherapy, which is assumed to be a suboptimal treatment for patients with a GC, was associated with weaker reduction in PLT, as shown in Figure 3. Differences in subgroups were formally tested with the Kruskal–Wallis H test that showed statistically significant differences between groups χ^2^(3) = 10.3, *p* = 0.0164.

To more fully reflect the inflammatory profile of the patients, controlling proportional changes in PLT for their initial values was needed. Both PLT-red and PLT-count were incorporated into the Cox regression analysis in the appropriate form, guaranteeing that there was no violation of the model assumptions. When we did not adjust to additional covariates, regardless of whether PLT-depl was included as a continuous or dichotomized variable, a high initial level of PLT was associated with approximately twice the risk of death (HR = 2.10, 95% CI: 1.30–3.38 or HR = 1.91, 95% CI: 1.19–3.06, respectively). Controlling for possible confounders showed an even stronger effect (HR = 2.32, 95% CI: 1.36–3.98 or HR = 2.29, 95% CI: 1.33–3.95, respectively). Regarding the time to progression of the disease, the estimates were also very stable, showing in fully adjusted models the tendency to a higher risk of outcome for patients with a PLT count greater than 400 × 10³/µL, however, with only marginally significant results (HR = 1.64, 95% CI: 0.99–2.73 or HR = 1.64, 95% CI: 0.98–2.76). We obtained a robust assessment of PLT on the impact of reduction on both OS and PFS with strong effect towards deterioration of prognosis across increasing values of PLT-red. The double increase in PLT-red was accompanied by a 60–66% higher risk of reaching OS/PFS endpoints (HR = 1.60, 95% CI: 1.12–2.31 for OS and HR = 1.66, 95% CI: 1.15–2.39 for PFS in fully adjusted models), and reaching at least a level of 0.6 was associated with a 55%–68% risk of OS and a 60–80% higher risk of the PFS endpoint (HR = 1.68, 95% CI: 1.06–2.64 for OS and HR = 1.80, 95% CI: 1.12–2.91 for PFS) (Table 2).

To translate these results into clinical practice, a final model with four groups determined by cross-classification of level of PLT-red (<60 and the ≥60) and concentration of PLT-count (<400 × 10³/µL and ≥400 × 10³/µL) was constructed and analyzed with Cox regression. Our goal was to check whether achieving a significant reduction in platelets could be beneficial even in the case of an elevated inflammatory reaction (Table 3).

In all constructed models, a decreasing trend of survival rates was detected across premade groups (*p* for trend < 0.05). After controlling for known confounders, the patient in the worst group (high PLT count and non-reduced PLT) had a 3.60 and 2.97 times greater risk of the death and progression of disease, respectively, in comparison to patients with optimal parameters (HR = 3.60, 95% CI: 1.41–9.18 for OS and HR = 2.97, 95% CI: 1.15–7.66 for PFS). Achieving platelet reduction in the group with a high inflammatory reaction improved their survival rates compared to patients with high inflammation and no platelet reduction. Differences in survival rates are visualized in Figure 4.

## 4. Discussion

Our analysis confirmed the results of some earlier studies indicating that thrombocytosis at the start of systemic treatment represents a detrimental prognostic factor in advanced gastrointestinal cancer [17], as well as other malignancies [18]. Chronic systemic inflammation mediated by cancer-induced cytokines (IL-6, IL-10) is reflected in the increase in the number of circulating platelets [19,20]. There is no doubt that platelets are not only a biomarker of advanced disease, but also a direct trigger of life-threatening conditions such as venous thromboembolism (VTE) [21] which, after neutropenia-induced infections, is the second leading cause of mortality in advanced malignancies [22]. Among various explanations for the detrimental impact of thrombocytosis on cancer patients’ outcomes, three appear to be the most important: (i) promotion of tumor growth and angiogenesis by PLT-secreted cytokines [23]; (ii) facilitation of metastasis by protecting circulating tumor cells from physical factors such as shear stress and host immune response [24]; and (iii) induction of a vicious circle in which platelets stimulate tumor cells, which further stimulate platelet production and activity [25].

Theoretically, normalization of an increased PLT count during cancer treatment should decrease the risk of VTE and improve patient outcomes [26]. However, robust data on the prognostic role of treatment-mediated mitigation of thrombocytosis in cancer patients are missing.

The observed impact of platelet reduction on survival could also be explained by the sheer effect of cytotoxic chemotherapy on malignant cells. In this setting, platelet reduction could reflect the intensity of myelosuppression induced directly by cytotoxic agents. Consequently, the lower decrease in platelets revealed by one-drug regimens supports this hypothesis.

In our opinion, it is implausible that only one of these explanations is correct. Platelet reduction directly inhibits the inflammation-induced tumor proliferation and indirectly reflect the antiproliferative potential of chemotherapy.

In clinical practice, the efficacy of systemic treatment is usually verified after 3 months of treatment. Therefore, we decided to analyze the extent of reduction in platelet count during the first 12 weeks of chemotherapy in the population studied. Based on initial statistical analyses, which revealed a survival curve plateau after reaching a 60% reduction in the PLT count, we decided to use this value as a cut-off point for further statistical analyzes.

Our results demonstrate that a significant reduction (60%) in cancer-related thrombocytosis represents a favorable prognostic factor for overall survival in chemotherapy-treated GC patients. The extent of reduction in PLT depends on the antitumor activity of chemotherapy and the specific toxicity of PLT of particular cytotoxic drugs such as carboplatin or gemcitabine [27]. However, in the systemic treatment of GC, none of the thrombogenesis-impacting medications mentioned above are used. Analysis of various chemotherapy regimens used in the population of patients studied revealed that multidrug (2-, 3-, trastuzumab-based regimens) regimens were significantly more effective in reducing PLT counts than monotherapy. Several studies and available treatment guidelines suggest the use of multidrug regimens (at least two drugs) in the first-line treatment of advanced GC, due to its supreme efficacy in terms of tumor response and patient outcomes [28,29]. Furthermore, our study confirmed that such multidrug regimens, which are associated with higher rates of objective responses, also efficiently inhibit tumor-related thrombocytosis and thus neutralize protumorigenic PLT-mediated mileau. What is also very important is that it was the combination of drugs, but not the intensity of chemotherapy, that resulted in a significant reduction in thrombocytosis and a subsequent improvement in overall survival. This observation suggests that in patients with GC who experience unacceptable treatment-related toxicity, it is better to continue multidrug-based chemotherapy at reduced doses than to switch to monotherapy.

Despite the advent of novel therapeutic approaches, advanced gastric cancer patients have a poor prognosis with a median overall survival that usually does not exceed 12 months [30]. Adopting new predictive factors that could optimize the early stages of cancer treatment could significantly benefit patients. The most profound platelets reduction was usually detected in the 7th week of treatment and therefore can act as a timely signal for early intervention.

Treatment of advanced gastric cancer is rapidly evolving due to the introduction of immune checkpoint inhibitors. Anti-PD1 inhibitors alone and in combination with chemotherapy have significant activity in advanced GC and will soon be approved as first- or second-line therapies in this indication [31]. However, many patients with GC will not respond to this approach and some of them may even experience a dangerous clinical phenomenon called hyperprogression, which is associated with extremely rapid disease progression and fatal outcomes. Recent publications analyze hyperprogression events in patients with various cancers [32], including gastrointestinal cancers [33]. They clearly showed that baseline thrombocytosis is a risk factor for this phenomenon.

Our study has several limitations. The most important limitation is its retrospective character. Furthermore, over the analyzed time, patients have been treated with various chemotherapy regimens based on the patient’s performance status and available guidelines. Therefore, the results should be validated prospectively to consider utilization of the thrombocytosis reduction as a marker to guide clinical decisions. Over the analyzed time, the recommended systemic treatment in advanced GC changed from the three-drug to two-drug regimens. In terms of overall survival, both three-drug and two-drug regimens are equal [34], therefore, it should not have had an impact on the result of our analysis. On the other hand, some bias could be caused by the selection of patients for one-drug regimes. This type of treatment has not been a standard of care in the analyzed period and has usually been reserved for patients in suboptimal general conditions.

The second limitation is the small study sample caused by the lack of longitudinal data for 50 of our patients. Although the dropout was considerable, the incompleteness of the data was random and should not give additional bias.

Consequently, our observations may provide some additional hints in clinical practice in advanced GC patients treated with standard chemotherapy-only regimens, but also in those treated with novel chemoimmunotherapies. However, as previously stressed, prospective validation of the clinical utility of thrombocytosis-reduction must be performed not only in chemotherapy-only but also in advanced GC treated with chemoimmunotherapy.

## 5. Conclusions

The degree of platelet reduction during the first 12 weeks of chemotherapy represents an important prognostic and predictive factor that affects both OS and PFS in advanced gastric cancer.

## Figures and Tables

**Figure 1 jcm-11-01213-f001:**
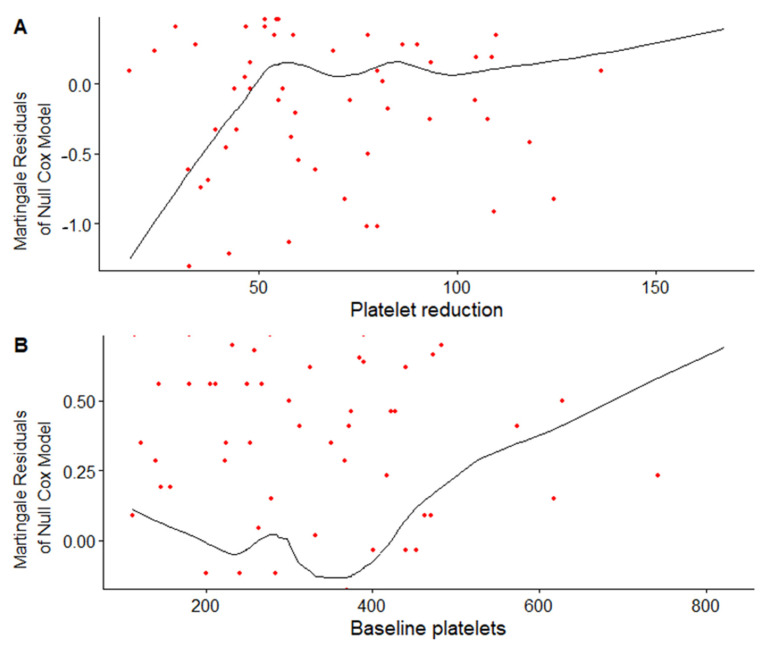
Relationship between PLT-red, PLT-count, and overall survival. (**A**) Martingale residuals of the null Cox model plotted against PLT-red. (**B**) Martingale residuals of Null Cox model plotted against PLT-count. The superimposed smooth line shows the approximation of the true functional form of a given covariate. The increasing trend shows shorter OS.

**Figure 2 jcm-11-01213-f002:**
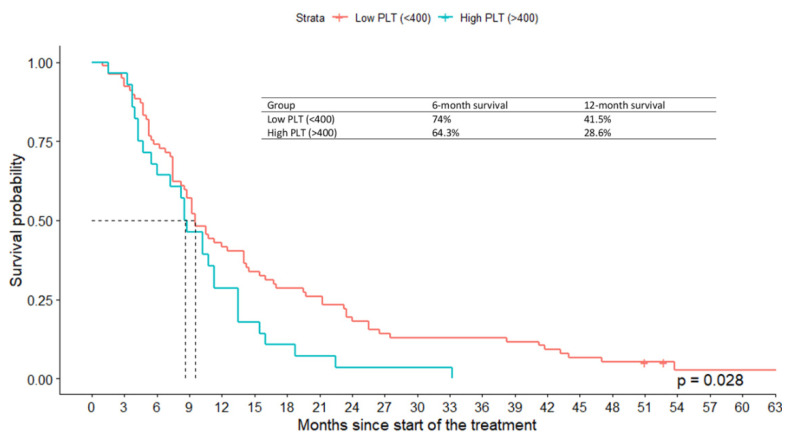
Overall survival according to the PLT-count in the high (PLT < 400 × 10³/µL) vs. low (PLT > 400 × 10³/µL) groups. Baseline patient demographic and clinical characteristics across reduced platelets (PLT-red < 60%) and non-reduced platelets (PLT-red 60%) patients are shown in Table 1.

**Figure 3 jcm-11-01213-f003:**
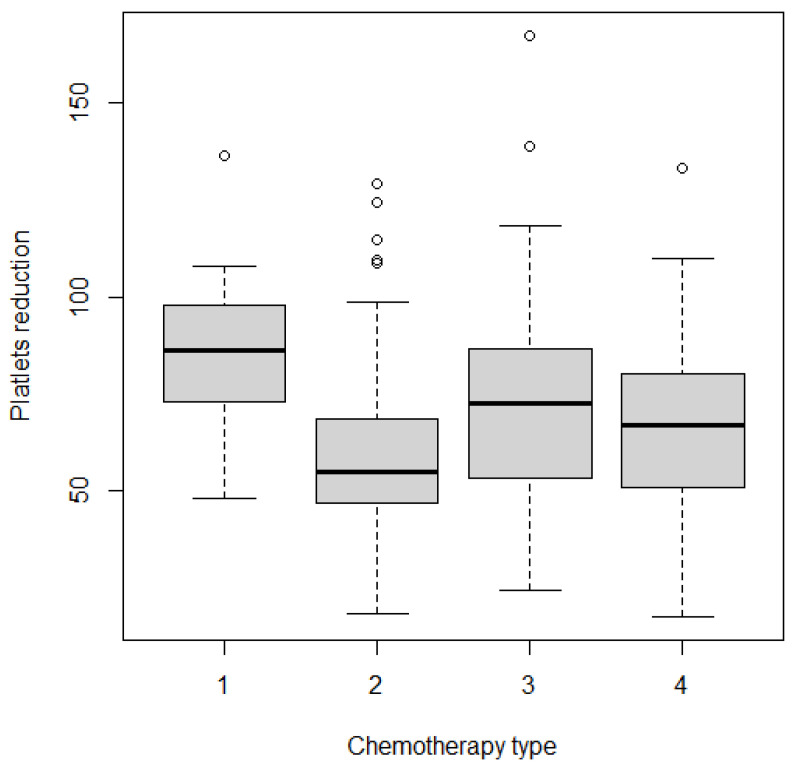
Comparison of platelet reduction grouped by type of chemotherapy administered. 1 = 1-drug regime, 2 = 2-drug regime, 3 = 3-drug regime, 4 = trastuzumab-based regime. ° = outliers.

**Figure 4 jcm-11-01213-f004:**
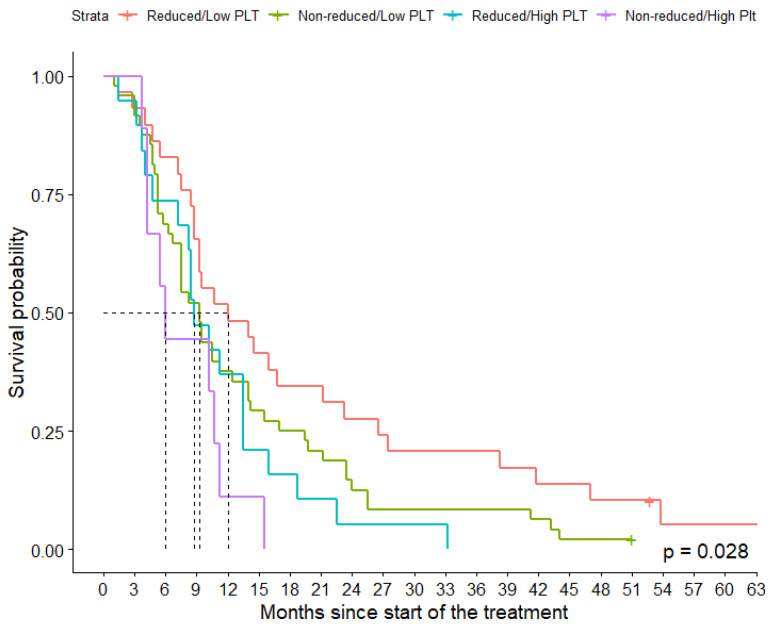
Kaplan–Meier curves for groups based on baseline platelet level and platelet reduction.

**Table 1 jcm-11-01213-t001:** Baseline characteristics.

	Reduced	Non-Reduced	
Variable	(*n* = 48)	(*n* = 57)	*p*
Age			
Median (Q1–Q3)	60.0 (54.8–70.3)	61.0 (54.0–70.0)	0.745
[Min, Max]	[32.0, 82.0]	[32.0, 82.0]	
Gender			
Male	32 (66.7%)	39 (68.4%)	1.000
Female	16 (33.3%)	18 (31.6%)	
Performance status(PS)			
0	8 (16.7%)	9 (15.8%)	0.967
1	30 (62.5%)	37 (64.9%)	
2	10 (20.8%)	11 (19.3%)	
Overall survival ‡			
Median (Q1–Q3)	297.5 (213.3–543.0)	262.0 (149.0–434.0)	0.171
[Min, Max]	[44.0, 1913.0]	[29.0, 1428.0]	
Progresion free survival ‡			
Median (Q1–Q3)	187.5 (118.8–295.0)	133.0 (81.0–233.0)	0.058
[Min, Max]	[44.0, 1477.0]	[29.0, 880.0]	
Platelets ¤			
Median (Q1–Q3)	377.5 (287.3–441.5)	259.0 (201.0–377.0)	0.000
[Min, Max]	[224.0, 822.0]	[113.0, 618.0]	
NEU ¤			
Median (Q1–Q3)	5.7 (4.2–6.4)	4.9 (3.8–6.0)	0.153
[Min, Max]	[1.9, 12.9]	[1.5, 10.1]	
LYM ¤			
Median (Q1–Q3)	1.5 (1.2–1.9)	1.6 (1.2–1.9)	0.561
[Min, Max]	[0.3, 2.8]	[0.7, 3.8]	
NLR			
Median (Q1–Q3)	3.6 (2.4–5.5)	3.2 (2.0–4.5)	0.129
[Min, Max]	[0.8, 15.0]	[1.0, 7.2]	
Missing	1 (2.1%)	1 (1.8%)	
PDW ±			
Median (Q1–Q3)	10.9 (10.3–12.4)	12.0 (10.7–13.6)	0.047
[Min, Max]	[8.8, 16.3]	[8.8, 18.7]	
Missing	1 (2.1%)	1 (1.8%)	
Chemotherapy type			
One-drug regime	2 (4.2%)	10 (17.5%)	0.043
Two-drugs regime	27 (56.3%)	19 (33.3%)	
Three-drugs regime	16 (33.3%)	21 (36.8%)	
Trastuzumab based regime	3 (6.3%)	7 (12.3%)	
Dose intensity			
Median (Q1–Q3)	0.81 (0.72–0.89)	0.81 (0.71–0.88)	0.634
[Min, Max]	[0.21, 1.00]	[0.47, 1.33]	

NEU = neutrophiles, LYM = lymphocytes, NLR = neutrophil to lymphocytes ratio, PDW = platelet distribution width, dose intensity = delivered cycles of chemotherapy/planned cycles. *p* = *p*-value of the Wilcoxon rank sum test for the difference between medians. ¤ = ×10³/µL. ± = fL, ‡ = days.

**Table 2 jcm-11-01213-t002:** Cox regression analysis for baseline platelet and platelets reduction.

		OS				PFS		
	Model 1 *		Model 2 **		Model 1 *		Model 2 **	
Inflammatory Markers	HR (95% CI)	*p*	HR (95% CI)	*p*	HR (95% CI)	*p*	HR (95% CI)	*p*
*Dichotomized PLT-count and logarithmically transformed PLT-red (continous)*								
PLT-count < 400 ¤	1.00 (ref.)	-	1.00 (ref.)	-	1.00 (ref.)	-	1.00 (ref.)	-
PLT-count ≥ 400 ¤	2.10	0.002	2.32 (1.36; 3.98)	0.002	1.56	0.061	1.64 (0.99; 2.73)	0.055
	(1.30; 3.38)				(0.98; 2.48)			
PLT-red (log2)	1.64	0.003	1.60 (1.12; 2.31)	0.011	1.62	0.004	1.66 (1.15; 2.39)	0.006
	(1.18; 2.28)				(1.16; 2.26)			
*Dichotomised both PLT-count and PLT-red*								
PLT-count < 400 ¤	1.00 (ref.)	-	1.00 (ref.)	-	1.00 (ref.)	-	1.00 (ref.)	-
PLT-count ≥ 400 ¤	1.91	0.008	2.29 (1.33; 3.95)	0.003	1.46	0.109	1.64 (0.98; 2.76)	0.060
	(1.19; 3.06)				(0.92; 2.32)			
PLT-red < 0.6	1.00 (ref.)	-	1.00 (ref.)	-	1.00 (ref.)	-	1.00 (ref.)	-
PLT-red > 0.6	1.55	0.038	1.68 (1.06; 2.64)	0.026	1.60	0.028	1.80 (1.12; 2.91)	0.015
	(1.02; 2.34)				(1.05; 2.42)			

* Model 1 includes only PLT-count and PLT-red. ** Model 2 includes PLT-count, PLT-red, ECOG, dose intensity, type of chemotherapy, and NLR at baseline (dichotomized with highest quartile as cut-off point). ¤ = × 10³/µL.

**Table 3 jcm-11-01213-t003:** Cox regression analysis for groups based on the baseline platelet level and platelet reduction.

			OS				PFS	
	Model 1 *		Model 2 **		Model 1 *		Model 2 **	
Inflammatory Markers	HR (95% CI)	*p*	HR (95% CI)	*p*	HR (95% CI)	*p*	HR (95% CI)	*p*
PLT-count <400 ¤ &	1.00 (ref.)	-	1.00 (ref.)	-	1.00 (ref.)	-	1.00 (ref.)	-
PLT-red < 0.6								
PLT-count <400 ¤ &	1.58 (0.97; 2.57)	0.064	1.75 (1.04; 2.93)	0.035	1.62 (0.99; 2.64)	0.055	1.80 (1.07; 3.05)	0.028
PLT-red ≥ 0.6								
PLT-count ≥400 ¤ &	1.97 (1.07; 3.61)	0.029	2.44 (1.26; 4.73)	0.008	1.49 (0.82; 2.72)	0.195	1.64 (0.87; 3.11)	0.127
PLT-red < 0.6								
PLT-count ≥400 ¤ &	2.89 (1.32; 6.33)	0.008	3.60 (1.41; 9.18)	0.007	2.30 (1.06; 4.99)	0.036	2.97 (1.15; 7.66)	0.024
PLT-red ≥ 0.6								
*p* for trend	0.003		0.001		0.041		0.028	

* Model 1 includes only PLT-count and PLT-red; ** Model 2 includes PLT-count, PLT-red, ECOG, dose intensity, type of chemotherapy, and NLR at baseline (dichotomized with highest quartile as a cut-off point). ¤ = ×10³/µL.

## Data Availability

The data presented in this study are available on request from the corresponding author. The data are not publicly available due to privacy concerns.

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
