# Peer review of "Reduction of Cancer-Induced Thrombocytosis as a Biomarker of Improved Outcomes in Advanced Gastric Cancer"

_jcm, 2022, doi:10.3390/jcm11051213_

Round 1

Reviewer 1 Report

The manuscript entitled “Reduction of cancer-induced thrombocytosis as a biomarker of improved outcomes in advanced gastric cancer” describes that interplay between non-specific inflammatory reaction and tumor microen-vironment in gastric cancer (GC) could be measured indirectly by assessing fluctuations in concen tration of platelets. Cytotoxic chemotherapy affects these morphotic elements directly by inducing myelosuppression. It is hypothesized that chemotherapy not only directly affects malignant cells, but also through myelosuppression-related immunomodulation. Authors found that degree of platelet reduction during first 12 weeks of chemotherapy is an important prognostic and predictive factor affecting both OS and PFS in gastric cancer.

  1. There are many similar research and review articles publicated in PubMed.
  2. Is the phenomenon specified for gastric cancer? Do other cancer types have the similar phenomenon about cancer-induced thrombocytosis?

Author Response

Thank you very much for your valuable review.

There are indeed many similar articles published on this topic, but this is caused by ongoing interest in immunotherapy as an important part of modern oncology. Recent studies have shown that targeted immunotherapy (PD-1, CTLA-4) is probably the most important invention in oncology in the last 50 years. Knowing how important is the immune response to cancer, multiple researchers are looking for similar connections with respect to classical cytotoxic chemotherapy, because this type of treatment will be relevant for many more years.

Regarding your second question, you are 100% right and I added relevant information and citation to the revised manuscript.  

Kamil Konopka

Reviewer 2 Report

Dear Editor, thank you so much for inviting me to revise this manuscript.

This study addresses a current topic.

The manuscript is quite well written and organized. English could be improved.

Figures and tables are comprehensive and clear.

The introduction explains in a clear and coherent manner the background of this study.

We suggest the following modifications:

  • Introduction section: although the authors correctly included important papers in this setting, we believe some studies should be cited within the introduction ( PMID: 33916206 ; PMID: 33916915 ), only for a matter of consistency. We think it might be useful to introduce the topic of this interesting study.
  • Methods and Statistical Analysis: nothing to add.
  • Discussion section: Very interesting and timely discussion. Of note, the authors should expand the Discussion section, including a more personal perspective to reflect on. For example, they could answer the following questions – in order to facilitate the understanding of this complex topic to readers: what potential does this study hold? What are the knowledge gaps and how do researchers tackle them? How do you see this area unfolding in the next 5 years? We think it would be extremely interesting for the readers.

However, we think the authors should be acknowledged for their work. In fact, they correctly addressed an important topic, the methods sound good and their discussion is well balanced.

One additional little flaw: the authors could better explain the limitations of their work, in the last part of the Discussion.

We believe this article is suitable for publication in the journal although some revisions are needed. The main strengths of this paper are that it addresses an interesting and very timely question and provides a clear answer, with some limitations.

We suggest a linguistic revision and the addition of some references for a matter of consistency. Moreover, the authors should better clarify some points.

Author Response

 Thank you very much for your valuable review.

All your comments were very useful and I think I responded to all of them in the revised manuscript.

I have tried to include answers to all your questions in the discussion and I hope you find it satisfying. There was indeed a problem with the study limitations section and I have practically rewritten all of it.  

 The proposed citation was also very valuable and was added to the bibliography.

Round 2

Reviewer 1 Report

Authors have well revised this manuscript according to reviewers' suggestions.

Reviewer 2 Report

The authors modified the manuscript according to our suggestions.

We recommend Acceptance.

This manuscript is a resubmission of an earlier submission. The following is a list of the peer review reports and author responses from that submission.